# RETHINKING DATA AUGMENTATION FOR ADVERSARIAL DISTILLATION: AN EXCESS RISK PERSPECTIVE

## ABSTRACT

Adversarial Robustness Distillation (ARD) enhances the robustness of lightweight models by transferring knowledge from robust teacher models. Most studies focus on output alignment, while input-side augmentation remains underexplored. We reveal a surprising phenomenon: augmentation techniques such as CutMix and AutoAugment, which work well in standard Knowledge Distillation (KD), are ineffective in ARD and can even reduce student robustness. To explain this, we derive an excess risk bound for ARD based on uniform stability, revealing how augmentation diversity and teacher performance on augmented data jointly affect generalization. Our analysis shows that, while augmentation improves sample diversity and smooths the loss landscape, low-quality or overly strong augmentations can compromise teacher reliability during training. This insight highlights a fundamental trade-off in ARD: effective augmentation must balance diversity with teacher reliability. To achieve this balance, we propose ASDA (Active Selection for Diffusion-based Augmentation), which leverages diffusion-generated samples and actively selects informative and teacher-reliable data, guided by output fidelity and entropy. Experiments on CIFAR-10/100 show that ASDA outperforms baselines and surpasses SOTA, clarifying the role of augmentation in ARD and providing a practical solution for improving student robustness.

## 1 INTRODUCTION

Deep learning models have achieved remarkable success in computer vision, natural language processing, and many other domains. Recent studies, however, have revealed their inherent security vulnerability: they are highly susceptible to adversarial examples. By adding imperceptible perturbations to the input, such attacks can mislead a model into making incorrect predictions. Adversarial Training (AT) (Szegedy et al., 2013), currently the most practical and effective defense, trains models directly on adversarial examples to substantially improve robustness. Although AT performs well on large models, its effectiveness degrades markedly for lightweight models deployed on resource-constrained edge devices, exposing these models to significant security risks.

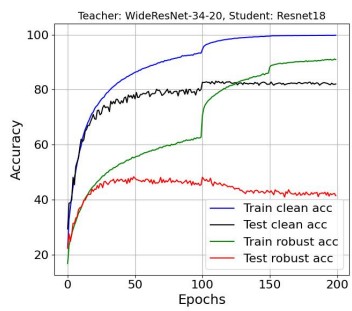

Figure 1: Robust Overfitting in ARD (Goldblum et al., 2020).

To address this challenge, Adversarial Robustness Distillation (ARD) has been proposed. ARD leverages a strongly adversarially trained teacher to guide the training of a smaller student, thereby improving the student's standard accuracy and adversarial robustness. However, most existing work emphasizes output-side knowledge alignment, for example by designing refined loss functions to better match teacher and student, while input-side strategies, i.e., data augmentation that acts directly on the training data, remain underexplored. This gap is crucial because ARD training often suffers from robust overfitting and large generalization error (see Fig. 1). Although data augmentation is an effective tool to mitigate robust overfitting and improve generalization by increasing data diversity, its mechanism and applicability in ARD have not been systematically studied. Notably, our experiments show that classical augmentations which are effective in standard Knowledge Distillation (KD) (Wang et al., 2022), such as CutMix

Table 1: Data augment methods results on CIFAR-10.

| Augmentation Method | Clean Acc ↑ | FGSM ↑ | PGD-20 ↑ | CW2 ↑ | AutoAttack ↑ | T-WRA ↑ | T-OS (mean ± std) ↓ |
|---|---|---|---|---|---|---|---|
| Base | 84.59% | 60.94% | 53.41% | 52.46% | 50.58% | **81.51%** | **0.0932 ± 0.0030** |
| Cutout | 84.36% | 59.02% | 52.69% | 51.82% | 50.18% | 48.18% | 0.1313 ± 0.0080 |
| CutMix | 83.94% | 59.60% | 53.86% | 52.18% | 50.79% | 41.63% | 0.2124 ± 0.0636 |
| RandAugment | 83.85% | 59.50% | 53.54% | 52.24% | 50.37% | 36.01% | 0.1313 ± 0.0080 |
| AutoAugment | 83.91% | 58.80% | 52.63% | 51.16% | 49.62% | 35.31% | 0.1364 ± 0.0071 |
| Diffusion-300k | **85.09%** | **62.05%** | **55.68%** | **54.27%** | **52.71%** | 74.03% | 0.1250 ± 0.0039 |

(Yun et al., 2019) and AutoAugment (Cubuk et al., 2019), fail to improve and may even degrade student performance in ARD. This counterintuitive observation raises two key questions: (1) Why do augmentations that work in KD fail in ARD? (2) What augmentations are suitable for ARD?

To address these questions, we conduct a statistical analysis of ARD's Excess Risk to quantify the impact of data augmentation on model performance. We extend prior analyses of data-dependent excess risk bounds in adversarial training to the setting of adversarial distillation, thereby obtaining a bound tailored for ARD. The results show that augmentation diversity and teacher behavior on augmented data jointly influence generalization by affecting gradient variance during training and the curvature around the initial risk. Further analysis indicates that, although augmentation improves sample diversity and smooths the loss landscape, low-quality or overly strong augmentations can undermine the teacher's reliability during training and ultimately degrade performance.

Building on this, we approximate the error constraint at $\epsilon = 0$ (i.e., standard KD) and compare the excess-risk bounds of KD and ARD, thereby addressing the first question. The results show that, compared to $\epsilon = 0$ (the KD setting), when $\epsilon > 0$ (the ARD setting), adversarial perturbations make the teacher's outputs more unstable and less accurate, thereby amplifying its unreliability and further enlarging the excess-risk bound. For example, as shown in Table 1, augmentations (including CutMix (Yun et al., 2019), Randaugment (Cubuk et al., 2020), Cutout (DeVries & Taylor, 2017), Diffusion-based augment (Wang et al., 2023), and AutoAugment (Cubuk et al., 2019)) significantly increase teacher-output instability (T-OS). Here, the base setting denotes only basic augmentations (random flip and random crop), while diffusion-300k refers to augmentation with 300k randomly generated diffusion samples. In particular, the instability of CutMix more than doubles relative to this baseline, and the teacher's classification accuracy on adversarial student examples (T-WRA) decreases by 40–46%.

Building on these analyses, we propose ASDA (Active Selection for Diffusion-based Augmentation), a framework designed to improve augmentation quality in ARD through active selection. The central idea is to use diffusion-generated samples as a diverse candidate pool and apply teacher-output–guided criteria to select samples that are both reliable and informative, thereby jointly accounting for data diversity and teacher reliability in practice.

Extensive experiments on CIFAR-10 and CIFAR-100 demonstrate that ASDA not only significantly outperforms existing heuristic augmentation methods but also surpasses current SOTA baselines. Our work clarifies, theoretically, how augmentation operates in ARD, and empirically shows that actively managing the trade-off between data diversity and the quality of the teacher's supervisory signal is an effective path to improving student adversarial robustness.

## 2 RELATED WORK

### 2.1 ADVERSARIAL DISTILLATION

Adversarial Robustness Distillation (ARD) aims to transfer knowledge from large, robust teacher models to compact student models, achieving both robustness and efficiency (Goldblum et al., 2020; Kuang et al., 2023; Lee et al., 2023; Zhao et al., 2022; Dong et al., 2024; Huang et al., 2023; Zhu et al., 2021; Jung et al., 2024; Zi et al., 2021; Yin et al., 2024). Since its inception, research has predominantly focused on optimization-centric strategies. For instance, Goldblum et al. (2020) pioneered the application of knowledge distillation to adversarial robustness, while subsequent works like RSLAD (Zi et al., 2021) and IAD (Zhu et al., 2021) designed sophisticated loss functions to balance standard and robust accuracy. Another line of research enhances knowledge transfer by

aligning intermediate layer features (Kuang et al., 2023) or attention maps (Yin et al., 2024) between teacher and student models. Recent studies, such as MTARD (Zhao et al., 2022), introduce multi-teacher mechanisms to mitigate issues like catastrophic forgetting. Despite these advances, a systematic blind spot persists: the role of input strategies, particularly data augmentation, is largely overlooked. Existing literature either omits data augmentation or relies on simplistic methods like random cropping and flipping, unaware of the risks posed by improper augmentation. In contrast, our work shifts the focus from "how to optimize distillation" to "how to optimize distillation inputs," revealing the complexity and critical importance of data augmentation in this unique framework.

## 2.2 ROBUST GENERALIZATION

Robust overfitting refers to the phenomenon where models, during adversarial training or distillation, overfit to specific adversarial perturbations, resulting in poor generalization to stronger or unseen attacks. Szegedy et al. (2013) first exposed the vulnerability of deep networks to adversarial examples, prompting Madry et al. (2017) to propose adversarial training, which enhances robustness by incorporating adversarial examples during training. However, Rice et al. (2020) demonstrated that adversarial training can lead to overfitting to specific attacks, reducing robustness against other perturbations. TRADES (Zhang et al., 2019) further mitigates overfitting by balancing robustness and accuracy through a tailored loss function. In terms of theoretical analysis, Bousquet & Elisseeff (2002) were the first to employ uniform stability to analyze the generalization error of adversarial training. Building on this, Xiao et al. (2022) introduced approximate gradient continuity to extend the theory to less smooth loss functions, and used the resulting bounds to explain overfitting. Furthermore, Wang et al. (2025) proposed on-average stability, which leads to data-dependent generalization constraints. Nevertheless, the impact of data augmentation on robust overfitting in adversarial distillation remains underexplored. Our work is the first to bridge this theoretical gap.

## 2.3 DATA AUGMENTATION

Data Augmentation (DA) is a core technique for enhancing model generalization by increasing the diversity of training data. In standard training, heuristic-based methods, exemplified by CutMix (Yun et al., 2019), and AutoAugment (Cubuk et al., 2019), are widely popular due to their strong regularization effects. However, their effectiveness in Adversarial Training (AT) is highly uncertain. Some studies, such as (Rebuffi et al., 2021), have shown that judiciously chosen augmentation strategies can work in synergy with AT to improve robustness. More recently, augmentation methods based on generative models, such as GANs (Antoniou et al., 2017), and Diffusion Models (Wang et al., 2023), have garnered attention. For instance, Wang et al. (2023) found that samples generated by Diffusion Models effectively enhance AT performance by providing stronger diversity compared to heuristic methods like CutMix (Yun et al., 2019). Nevertheless, these discussions on data augmentation are almost exclusively confined to the context of standard adversarial training. However, these analyses rarely address the dimension of ARD. Consequently, a critical research gap emerges at the intersection of these fields, motivating our work: while DA has been studied separately in both AT and standard KD, its role and underlying mechanisms in the unique context of ARD remain a systematically un-analyzed domain. Our work is the first to dissect this issue from a statistical perspective, revealing the root causes for why many popular DA strategies fail in ARD.

## 3 GENERALIZATION CHALLENGES IN ADVERSARIAL ROBUSTNESS DISTILLATION

In Adversarial Robustness Distillation (Goldblum et al., 2020), a large robustness generalization gap is observed: as shown in Fig. 1, although training robustness can reach above 90%, test robustness may drop to around 40%, revealing a severe failure to generalize. This raises the question of whether data augmentation, which is effective in standard Knowledge Distillation, can mitigate such a gap. However, as reported in Table 1, experiments on CIFAR-10 with WRN-34-10 as teacher and ResNet-18 as student show that common augmentations such as CutMix (Yun et al., 2019), Cutout (DeVries & Taylor, 2017), AutoAugment (Cubuk et al., 2019), and RandAugment (Cubuk et al., 2020) do not improve robustness and sometimes even degrade performance, whereas diffusion-based augmentations (Wang et al., 2023) consistently outperform the baseline (training with only basic augmentation: random flip and random crop) across all robustness metrics. In ARD, we track two

teacher-related metrics: T-WRA (during training, the teacher's worst-case robust accuracy on the training adversarial samples) and T-OS (the KL divergence between teacher outputs on clean inputs and their adversarial counterparts, $\mathrm{KL}\big(f_T(x) \,\|\, f_T(x')\big)$). T-OS is computed on 1,000 randomly sampled augmentations, averaged over ten runs, and reported as mean $\pm$ std. Under classical augmentations, we observe a sharp decrease in T-WRA and an increase in T-OS, indicating degraded teacher robustness; this explains why strong augmentations like CutMix succeed in KD but fail in ARD and motivates our theoretical analysis of when and how augmentation can improve robust generalization.

## 4    THEORETICAL ANALYSIS

In this section, we analyze the excess risk of adversarial robustness distillation and derive the corresponding upper bounds. We further investigate how data augmentation influences the key factors in these bounds.

### 4.1    PROBLEM SETUP

Adversarial robustness distillation leverages a robust teacher model to train the student, aiming to minimize the prediction discrepancy on both clean and adversarial inputs. Let $\mathcal{D}$ be the data distribution and $S = \{(x_i, y_i)\}_{i=1}^n \overset{\text{i.i.d.}}{\sim} \mathcal{D}^n$. Given a fixed robust teacher $f_T$ and a student $f_S(\theta; \cdot)$, We write the optimization objective (AdaAD-style (Huang et al., 2023)) as:

$$\min_\theta \; \mathbb{E}_{x \sim \mathcal{D}}\big[(1 - \alpha) \cdot \mathrm{KL}\big(f_T(x) \,\|\, f_S(\theta; x)\big) + \alpha \cdot \mathrm{KL}\big(f_T(x^\star) \,\|\, f_S(\theta; x^\star)\big)\big], \tag{1}$$

where $\alpha \in [0, 1]$ balances clean and adversarial alignment, $\|\cdot\|_p$ denotes the $\ell_p$-norm ($p \geq 1$), and $\varepsilon > 0$ is the adversarial budget.

Equivalently, the adversarial example is defined by

$$x^\star \in \arg \max_{\|x'-x\|_p \leq \varepsilon} \mathrm{KL}\big(f_T(x') \,\|\, f_S(\theta; x')\big). \tag{2}$$

In the appendix A.5, we show that the loss on clean samples does not alter the main conclusions of the subsequent analysis; hence, for simplicity, we set $\alpha = 1$ here. A special case arises when the teacher's outputs on adversarial examples are excluded from the loss; in this case, our analysis approximately treats the teacher output as a argmax hard label (Goldblum et al., 2020; Zi et al., 2021).

We then can simplify the ARD surrogate loss as:

$$h(\theta; x) \overset{\text{def}}{=} \max_{\|x'-x\|_p \leq \varepsilon} g(\theta; x'), \text{ where } g(\theta; x') = \ell(f_S(\theta; x'); f_T(x')). \tag{3}$$

Here, we use $\ell$ to denote a discrepancy (e.g., KL or cross-entropy).[1] Then, the ARD objective can be written as:

$$\min_\theta h(\theta; x) = \min_\theta \max_{\|x'-x\|_p \leq \varepsilon} g(\theta; x') = \min_\theta \max_{\|x'-x\|_p \leq \varepsilon} \ell(f_S(\theta; x'); f_T(x')) \tag{4}$$

The population risk $R_\mathcal{D}(\theta)$ and empirical distillation risk $R_S(\theta)$ are defined as:

$$R_\mathcal{D}(\theta) \overset{\text{def}}{=} \mathbb{E}_{(x,y) \sim \mathcal{D}} \, \ell(f_S(\theta; x'); y), \quad R_S(\theta) \overset{\text{def}}{=} \frac{1}{n} \sum_{i=1}^n h(\theta; x_i). \tag{5}$$

The population risk measures how well the student aligns with the true labels across the underlying data distribution, typically estimated using a held-out test set. The empirical distillation risk instead reflects how closely the student matches the teacher on the training set, corresponding to the average training loss. In addition, we introduce the complementary quantities:

$$\hat{R}_\mathcal{D}(\theta) \overset{\text{def}}{=} \mathbb{E}_{x \sim \mathcal{D}} \, h(\theta; x), \qquad \hat{R}_S(\theta) \overset{\text{def}}{=} \frac{1}{n} \sum_{i=1}^n \ell(f_S(\theta; x_i); y_i). \tag{6}$$

---

[1] We follow prior setups (Wang et al., 2022; 2025; Xiao et al., 2022). Cross-entropy and KL divergence are not globally smooth on the simplex, but both are twice differentiable in its interior ($q_i > 0$). Under controlled training with a uniformly bounded Hessian, they satisfy the same differentiability conditions.

Here, the $\hat{R}_{\mathcal{D}}(\theta)$ represents the student's predictive accuracy with respect to the true labels on the training set, while the $\hat{R}_S(\theta)$ reflects its alignment with the teacher over the data distribution. These definitions will be used for the decomposition of the excess risk of ARD.

**Excess Risk Decomposition.** The goal of training is to find a model $\hat{\theta}$ whose expected risk $R_D(\hat{\theta}; y)$ is as close as possible to that of the optimal model $\theta^*$. This gap, termed the excess risk, reveals the difference between a specific model's population risk and the population risk of the optimal model within its entire hypothesis space. Let $\theta^* = \arg\min_\theta R_D(\theta; y)$ and $\bar{\theta} = \arg\min_\theta R_S(\theta; f_T)$. Based on these definitions, the excess risk in robustness distillation can be decomposed as follows:

$$\underbrace{R_D(\hat{\theta}) - R_D(\theta^*)}_{\text{Excess Risk}} = \underbrace{R_D(\hat{\theta}) - R_S(\hat{\theta})}_{\mathcal{E}_{\text{gen}}} + \underbrace{R_S(\hat{\theta}) - R_S(\bar{\theta})}_{\mathcal{E}_{\text{opt}}}$$
$$+ \underbrace{R_S(\bar{\theta}) - R_S(\theta^*)}_{(1)\ \leq 0} + \underbrace{R_S(\theta^*) - R_D(\theta^*)}_{(2)\ E=0} + \underbrace{R_D(\theta^*) - R_D(\theta^*)}_{(3)\ \text{Const}}. \tag{7}$$

In the above decomposition, the last three terms need no further discussion. The first term is always non-positive because $\bar{\theta} = \arg\min_\theta R_S(\theta; f_T)$. The second term vanishes since $\theta^* = \arg\min_\theta R_D(\theta; y)$ and $S$ is a random sample from $\mathcal{D}$, implying $\mathbb{E}_S[R_S(\theta; f_T)] = R_D(\theta; f_T)$. The third term is a constant because $\theta^*$ is fixed and independent of training. Theory and empirical evidence show that the robust-optimization gap in adversarial training is typically small (Nemirovski et al., 2009a;b; Wu et al., 2020). Consequently, in adversarial distillation the excess risk is dominated by generalization. As Fig. 1 illustrates, the generalization term is dominant, so we focus on how data shapes the generalization bound.

## 4.2 DATA-DEPENDENT STABILITY IN ADVERSARIAL ROBUSTNESS DISTILLATION

A widely used and powerful framework for analyzing the generalization gap is algorithmic stability (Bousquet & Elisseeff, 2002; Xiao et al., 2022). Formally, we consider the notion of uniform stability.

**Definition 4.1** (Uniform Stability). *A randomized algorithm $\mathcal{A}$ is defined as $\epsilon$-uniformly stable if for all data sets $S, S' \in \mathcal{X}^n$ that differ in at most one example, we have:*

$$\sup_z \mathbb{E}_{\mathcal{A}}[h(\mathcal{A}(S); x) - h(\mathcal{A}(S'); x)] \leq \epsilon. \tag{8}$$

**Theorem 4.2.** *Let an algorithm $\mathcal{A}$ is $\epsilon$-uniformly stable, then its expected generalization gap is bounded :*

$$|\mathbb{E}_{S,\mathcal{A}}[R_D(\mathcal{A}(S)) - R_S(\mathcal{A}(S))]| \leq \epsilon. \tag{9}$$

Uniform stability is a classic way to analyze generalization, requiring an algorithm to remain stable in the worst case over all data points. This yields universal but data-independent bounds, often too pessimistic and missing the stronger stability an algorithm may show on real datasets. To capture performance more practically, we need data-dependent stability that reflects the actual data distribution.

In order to analyze the data-dependent stability, we employ the notion of on-average stability (Wang et al., 2025). Given a dataset $S = \{x_1, \ldots, x_n\} \sim \mathcal{D}^n$ and $x \sim \mathcal{D}$, replacing $x_i$ in $S$ with $x$, we denote $S^{i,x} = \{x_1, \ldots, x_{i-1}, x, x_{i+1}, \ldots, x_n\}$ with $i \in [n]$.

**Definition 4.3** (On-Average Stability). *A randomized algorithm $\mathcal{A}$ is $\epsilon$-on-average stable if*

$$\sup_{i \in [n]} \mathbb{E}_{S,x,\mathcal{A}}[h(\mathcal{A}(S), x) - h(\mathcal{A}(S^{i,x}), x)] \leq \epsilon. \tag{10}$$

In contrast, on-average stability averages over the draw of both the training set $S$ and the replacement point $x$ from the true data distribution $\mathcal{D}$. This averaging property is crucial, as it avoids the stringent worst-case requirement and is key to deriving data-dependent generalization bounds.

We build our analyses on the following standard assumptions (Bousquet & Elisseeff, 2002; Xiao et al., 2022; Wang et al., 2025). We next introduce a set of assumptions to analyze the excess risk bounds of adversarial robustness distillation. Since the adversarial loss $h$ involves a maximization and thus lacks direct continuity, we first impose suitable continuity conditions on the underlying function $g$ inside $h$.

**Assumption 4.4.** *Assume $g$ is twice differentiable, and for all $\theta_1, \theta_2$ and $x \in \mathcal{X}$, the following conditions hold.*

*1. The function $g$ is $L$-Lipschitz in $\theta$ and $L_x$-Lipschitz in $x$:*

$$|g(\theta_1; x) - g(\theta_2; x)| \leq L\|\theta_1 - \theta_2\|, \tag{11}$$

$$|g(\theta; x_1) - g(\theta; x_2)| \leq L_x\|x_1 - x_2\|. \tag{12}$$

*2. The gradient $\nabla g$ is $L_\theta$-gradient Lipschitz in $\theta$ and $L_x$-gradient Lipschitz in $x$:*

$$\|\nabla g(\theta_1; x) - \nabla g(\theta_2; x)\| \leq L_\theta\|\theta_1 - \theta_2\|, \tag{13}$$

$$\|\nabla g(\theta; x_1) - \nabla g(\theta; x_2)\| \leq L_x\|x_1 - x_2\|. \tag{14}$$

*3. The Hessian $\nabla^2 g$ is $H_\theta$-Hessian Lipschitz in $\theta$ and $H_x$-Hessian Lipschitz in $x$:*

$$\|\nabla^2 g(\theta_1; x) - \nabla^2 g(\theta_2; x)\| \leq H_\theta\|\theta_1 - \theta_2\|, \tag{15}$$

$$\|\nabla^2 g(\theta; x_1) - \nabla^2 g(\theta; x_2)\| \leq H_x\|x_1 - x_2\|. \tag{16}$$

Since the teacher model $f_T$ is robustly pre-trained and kept fixed during distillation, we impose the following local stability and boundedness assumption.

**Assumption 4.5** (Teacher Adversarial local Lipschitz continuity)**.** *There exist constants $L_T > 0$ such that for any $x \sim \mathcal{D}$: The teacher model $f_T$ is $L_T$-Lipschitz continuous within the adversarial perturbation ball:*

$$\forall x_1', x_2' \in \mathcal{B}(x, \epsilon), \quad \|f_T(x_1') - f_T(x_2')\| \leq L_T \|x_1' - x_2'\|. \tag{17}$$

*If the augmented distribution reasonably approximates the true distribution, a constant $L_T$ ensuring Eq. 17 still exists, with its value determined by the teacher's stability on the augmented data. Conversely, if the training distribution is a poor approximation of the true one, the existence of such a bound can no longer be guaranteed.*

**Definition 4.6.** *Let $\eta, \beta, \nu, \rho > 0$ and $h(\theta)$ be a second-order differentiable function.*
*1. $h$ is $\eta$-approximately $\beta$-gradient Lipschitz with respect to $\theta$, if*

$$\|\nabla h(\theta_1) - \nabla h(\theta_2)\| \leq \beta\|\theta_1 - \theta_2\| + \eta. \tag{18}$$

*2. $h$ is $\nu$-approximately $\rho$-Hessian Lipschitz with respect to $\theta$, if*

$$\|\nabla^2 h(\theta_1) - \nabla^2 h(\theta_2)\| \leq \nu\|\theta_1 - \theta_2\| + \rho. \tag{19}$$

*When $h$ is convex in $\theta$, the analysis can be relaxed to require only first-order differentiability.*

Based on the above assumptions and definitions, we analyze the approximate gradient smoothness of the loss function $h$ with respect to $\theta$. From Assumption 4.4, Assumption 4.5 and Definition 4.6, we obtain the following lemma in the setting of ARD (appendix A.4).

**Lemma 4.7.** *Let $h$ be the adversarial loss defined in Eq. 4 and $g$ satisfies Assumption 4.4. For all $\theta_1, \theta_2$ and $\forall x \in \mathcal{X}$, the following properties hold.*
*1. $h$ is $L$-Lipschitz with respect to $\theta$:*

$$|h(\theta_1, x) - h(\theta_2, x)| \leq L\|\theta_1 - \theta_2\|. \tag{20}$$

*2. $h$ is $(L_z L_T + L_x) \cdot 2\varepsilon$-approximately $L_\theta$-gradient Lipschitz with respect to $\theta$:*

$$\|\nabla h(\theta_1, x) - \nabla h(\theta_2, x)\| \leq L_\theta\|\theta_1 - \theta_2\| + (L_z L_T + L_x) \cdot 2\varepsilon. \tag{21}$$

*3. $h$ is $(H_z L_T + H_x) \cdot 2\varepsilon$-approximately $H_\theta$-Hessian Lipschitz with respect to $\theta$:*

$$\|\nabla^2 h(\theta_1, x) - \nabla^2 h(\theta_2, x)\| \leq H_\theta\|\theta_1 - \theta_2\| + (H_z L_T + H_x) \cdot 2\varepsilon. \tag{22}$$

Having established the key Lipschitz-like properties of the adversarial loss function h in Lemma 4.7, we now possess the necessary technical tools for our main analysis. Our primary goal is to bound the generalization error of ARD under data augmentation. As a first step, we decompose this total generalization error into two more manageable components.

**Decomposition of ARD Generalization Error in Data Augmentation.** After applying the data augmentation transformation $\mathcal{T}(\cdot)$, the resulting distribution becomes $\mathcal{D}_\mathcal{T}$. In this case, the generalization error bound $\mathcal{E}_{gen}$ can be decomposed as:

$$\mathcal{E}_{gen} = \underbrace{R_\mathcal{D}(\hat{\theta}) - R_{\mathcal{D}_\mathcal{T}}(\hat{\theta})}_{\mathcal{E}_{gen,\mathcal{D}_\mathcal{T}}^{\text{Target}}} + \underbrace{R_{\mathcal{D}_\mathcal{T}}(\hat{\theta}) - R_{\mathcal{T}(S)}(\hat{\theta})}_{\mathcal{E}_{gen,\mathcal{D}_\mathcal{T}}^{\text{ARD}}}. \tag{23}$$

This decomposition consists of two parts: a generalization term ($\mathcal{E}_{gen,\mathcal{D}_\mathcal{T}}^{\text{ARD}}$) on the augmented distribution, and a bias term ($\mathcal{E}_{gen,\mathcal{D}_\mathcal{T}}^{\text{Target}}$) measuring the deviation between the augmented and target distributions. We now proceed to analyze the upper bounds of these two terms.

### 4.3 Analysis of Generalization in Data Augmentation

We first proceed to bound the bias term, $\mathcal{E}_{gen,\mathcal{D}_\mathcal{T}}^{\text{Target}}$.

**Theorem 4.8.** *Under Assumptions 4.4 and 4.5, the bias term is bounded for any adversarial example $\hat{x}' \in \mathcal{B}(\hat{x}, \varepsilon)$ as follows:*

$$\varepsilon_{gen,\mathcal{D}_\mathcal{T}}^{\text{Target}} = |R_\mathcal{D}(\hat{\theta}) - R_{\mathcal{D}_\mathcal{T}}(\hat{\theta})| \leq L_z \mathbb{E}_{\hat{x} \sim \mathcal{D}_\mathcal{T}}[\|f_T(\hat{x}') - y\|] \tag{24}$$

*where $\mathcal{B}(\hat{x}, \varepsilon) := \{\hat{x}' \in \mathcal{X} \mid \|\hat{x}' - \hat{x}\|_p \leq \varepsilon\}$, $p \geq 1$.*

Therefore, Theorem 4.8 shows that the bias term $\mathcal{E}_{gen,\mathcal{D}_\mathcal{T}}^{\text{Target}}$ is bounded by the teacher's expected error on the augmented distribution, $\mathbb{E}_{\hat{x} \sim \mathcal{D}_\mathcal{T}}[\|f_T(\hat{x}') - y\|]$. Detailed proof is provided in Appendix A.6. This term quantifies the bias from the teacher's error on the augmented distribution $\mathcal{D}_\mathcal{T}$. The bound depends only on the teacher $f_T$ and the distribution $\mathcal{D}_\mathcal{T}$, tightening as the teacher's accuracy on augmented data improves.

Next, we focus on $\mathcal{E}_{gen,\mathcal{D}_\mathcal{T}}^{\text{ARD}}$. We extend the prior theory on the impact of data on the generalization bound from adversarial training (Wang et al., 2025), aiming to establish a data-dependent generalization error upper bound within the adversarial robustness distillation framework. This bound is determined by the properties of the data distribution, along with the model's initialization point $\theta_1$.

**Theorem 4.9** (Data-Dependent Generalization Bound (Wang et al., 2025)). *Assume the adversarial loss $h$ is non-negative and satisfies Assumption 4.4, $f_T$ satisfies Assumption 4.5. Let the model be trained for $T$ iterations via SGD, $t$ is the $t$-th iteration, starting from an initial point $\theta_1$. Under different convexity assumptions, the generalization term $\mathcal{E}_{gen,\mathcal{D}_\mathcal{T}}^{\text{ARD}}$ is bounded as follows:*

***Convex Case.*** *If $h(\theta; x)$ is convex, $L$-Lipschitz, and $\eta$-approximately $\beta$-gradient Lipschitz in $\theta$, then using step size $\alpha_t \leq 1/\beta$, we have*

$$\mathcal{E}_{gen,\mathcal{D}_\mathcal{T}}^{ARD} \leq \left(2\sigma L + L\eta\right) \sum_{t=1}^T \alpha_t + \frac{4L}{\sqrt{n}} \sqrt{\sum_{t=1}^T \alpha_t \left(\mathcal{R}_{\mathcal{D}_\mathcal{T}}(\theta_1) - \mathcal{R}_{\mathcal{D}_\mathcal{T}}(\theta^*) + \frac{\beta\sigma^2}{2} \sum_{t=1}^T \alpha_t^2 + \eta L \sum_{t=1}^T \alpha_t\right)}. \tag{25}$$

***Non-Convex Case.*** *If $h(\theta; x)$ is $L$-Lipschitz, $\eta$-approximately $\beta$-gradient Lipschitz, and $\nu$-approximately $\rho$-Hessian Lipschitz in $\theta$, then with step size $\alpha_t = c/t$ ($c \leq \min\{\frac{1}{\beta}, \frac{1}{4\beta \ln T}, \frac{1}{8(\beta \ln T)^2}\}$), we have*

$$\mathcal{E}_{gen,\mathcal{D}_\mathcal{T}}^{ARD} \leq \frac{1 + \frac{1}{c\gamma}}{n} \left(2cL^2 + nc\eta L\right)^{\frac{1}{1+c\gamma}} \cdot \left(\mathbb{E}_{S,\mathcal{A}}[\mathcal{R}_{\mathcal{D}_\mathcal{T}}(\mathcal{A}(S))]^T\right)^{\frac{c\gamma}{1+c\gamma}}, \tag{26}$$

*where $\gamma = \min\{\beta, \tilde{\mathcal{O}}(\hat{H} + \nu + \Delta^*)\}$ and $\Delta^* = \rho(\sqrt{(\mathcal{R}_{\mathcal{D}_\mathcal{T}}(\theta_1) - \mathcal{R}_{\mathcal{D}_\mathcal{T}}(\theta^*))c} + c\sigma + c\sqrt{\eta L})$, $\sigma^2 = \sup_\theta \mathbb{E}_{\hat{x} \sim \mathcal{D}_\mathcal{T}}[\|\nabla h(\theta, x) - \nabla \mathcal{R}_{\mathcal{D}_\mathcal{T}}(\theta)\|^2]$, $\eta = (L_z L_T + L_x) \cdot 2\varepsilon$, $\nu = (H_z L_T + H_x) \cdot 2\varepsilon$, $\hat{H} = \mathbb{E}_x[\|\nabla^2 h(\theta_1, x)\|]$.*

This theorem provides an upper bound for the generalization error, denoted as $\mathcal{E}_{gen,\mathcal{D}_\mathcal{T}}^{\text{ARD}}$. Our analysis models the training process as stochastic optimization navigating a loss landscape. The geometric properties of this landscape are defined by its Lipschitz continuity ($L$), gradient smoothness ($\beta$), and Hessian smoothness ($\rho$). The optimization trajectory is further influenced by factors such as stochastic gradient variance ($\sigma^2$), the risk at initialization ($R_{\mathcal{D}_\mathcal{T}}(\theta_1)$), and the discrepancy between the initial risk and the optimal risk ($R_{\mathcal{D}_\mathcal{T}}(\theta^*)$).

In conventional adversarial training, critical constants related to the loss landscape, such as $\beta_{AT} = 2L_z\varepsilon$ and $\nu_{AT} = 2H_z\varepsilon$, are predominantly determined by the loss function itself, rather than the underlying data distribution. In this context, data primarily affects the optimization process through its influence on gradient variance, the curvature at the point of initialization, and the distance from initialization to the optimal solution. It's worth noting that considering the randomness of the initial point, smoothing the expected global loss gradient curvature can reduce the initial point's curvature term. However, the risk gap between the initial and optimal points is difficult to be directly controlled; therefore, our primary focus is on the former two terms.

Conversely, in Adversarial Robustness Distillation, the data distribution also indirectly modifies the loss landscape by impacting the stability of the teacher model. An unreliable teacher renders the landscape more turbulent and exacerbates the generalization error. Therefore, even if data augmentation helps to smooth the loss landscape (Chapelle et al., 2000) and simultaneously reduce both variance and expected curvature, an unstable or inaccurate teacher will still inflate both $\mathcal{E}_{gen,\mathcal{D}_\mathcal{T}}^{\text{ARD}}$ and $\mathcal{E}_{gen,\mathcal{D}_\mathcal{T}}^{\text{Target}}$. This ultimately results in a higher overall generalization error.

**Key Insight.** In ARD, data augmentation faces a trade-off between diversity and teacher reliability. Strong or low-quality augmentations may boost diversity but can undermine the teacher's stability and correctness in adversarial regions, thereby enlarging the generalization error. Thus, effective augmentation must balance these two factors.

## 5 ASDA: ACTIVE SELECTION FOR DIFFUSION-BASED AUGMENTATION

We begin with the standard setting of ARD, where the student model $f_\theta$ is trained to mimic the adversarially trained teacher $f_T$. Given a training set $S = \{(x_i, y_i)\}_{i=1}^n \sim D$, the adversarial distillation objective is

$$\min_\theta h(\theta; x_i, f_T) = \min_\theta KL\big(f_\theta(x_i'), f_T(x_i')\big), \quad x_i' = \arg \max_{x \in \mathcal{B}(x_i,\varepsilon)} CE(f_\theta(x), y). \quad (27)$$

Building on our theoretical analysis, we argue that the effectiveness of augmentation in ARD relies on balancing sample diversity and teacher reliability. To achieve this, we propose ASDA (Active Selection for Diffusion-based Augmentation), which employs diffusion models to generate a large candidate pool of augmented samples and then actively filters them using teacher-guided criteria.

Specifically, diversity is quantified by the entropy of the teacher's prediction

$$H_T(x') = -\sum_{k=1}^K p_T(y = k \mid x') \log p_T(y = k \mid x'). \quad (28)$$

While reliability is measured by the distance between the teacher's output and the ground-truth label

$$D_T(x') = \|f_T(x') - y\|_2. \quad (29)$$

Where higher entropy indicates more informative samples and smaller distance implies more reliable supervision. To combine these two factors, we define a unified scoring function

$$S(x') = \hat{\alpha} \cdot H_T(x') - (1 - \hat{\alpha}) \cdot D_T(x'), \quad (30)$$

with $\hat{\alpha} \in [0, 1]$ controlling the trade-off. During training, the highest-scoring samples are selected from the diffusion-generated pool to form the augmented dataset for ARD.

## 6 EXPERIMENTAL ANALYSES

### 6.1 RESULTS ON CIFAR-10 AND CIFAR-100

In Table 2, the distillation results on the classical benchmarks CIFAR-10 and CIFAR-100 demonstrate that our method consistently achieves superior performance across diverse scenarios. Whether the student model is ResNet-18 or MobileNet-V2, our method surpasses existing approaches in both clean accuracy and robustness under various adversarial attacks (FGSM (Goodfellow et al., 2014), PGD-20 (Madry et al., 2017), $CW_\infty$ (Carlini & Wagner, 2017), AutoAttack (Croce & Hein, 2020)).

Table 2: Results on CIFAR-10 and CIFAR-100.

| Model | Method | CIFAR-10 | | | | | CIFAR-100 | | | | |
|-------|--------|-----------|--------|---------|-----------|--------------|-----------|--------|---------|-----------|--------------|
| | | Clean Acc ↑ | FGSM ↑ | PGD-20 ↑ | CW2 ↑ | AutoAttack ↑ | Clean Acc ↑ | FGSM ↑ | PGD-20 ↑ | CW2 ↑ | AutoAttack ↑ |
| WRN-34-10 | Teacher | 84.92% | 61.17% | 55.29% | 53.96% | 52.91% | 64.07% | 39.09% | 36.16% | 32.05% | 30.85% |
| RN-18 | ARD (Goldblum et al., 2020) | 82.76% | 57.98% | 50.85% | 50.28% | 48.54% | 60.53% | 33.20% | 30.49% | 28.42% | 25.89% |
| | RSLAD (Zi et al., 2021) | 82.69% | 59.86% | 53.53% | 52.07% | 50.86% | 59.14% | 35.33% | 32.56% | 28.58% | 27.08% |
| | IAD (Zhu et al., 2021) | 82.32% | 57.74% | 51.84% | 50.44% | 49.12% | 57.82% | 34.90% | 31.93% | 28.02% | 26.52% |
| | AdaAD (Huang et al., 2023) | 83.85% | 58.23% | 52.88% | 50.37% | 48.69% | 62.84% | 36.30% | 33.17% | 29.61% | 28.16% |
| | PeerAiD (Jung et al., 2024) | 85.06% | 61.01% | 54.37% | 55.25% | 52.89% | 59.43% | 34.21% | 29.44% | 29.64% | 27.24% |
| | Ours (300k) | **85.38%** | **62.27%** | **56.08%** | **55.23%** | **53.63%** | **61.80%** | **36.83%** | **33.78%** | **30.14%** | **28.68%** |
| MN-V2 | ARD (Goldblum et al., 2020) | 82.14% | 55.97% | 49.74% | 48.90% | 47.37% | 57.11% | 30.21% | 28.88% | 26.21% | 24.43% |
| | RSLAD (Zi et al., 2021) | 82.61% | 58.98% | 53.13% | 51.66% | 50.54% | 59.41% | 34.83% | 32.14% | 28.77% | 26.97% |
| | IAD (Zhu et al., 2021) | 81.39% | 56.99% | 51.83% | 50.11% | 48.39% | 57.12% | 32.88% | 29.48% | 26.25% | 24.69% |
| | AdaAD (Huang et al., 2023) | 84.13% | 57.75% | 52.32% | 51.00% | 49.93% | 62.91% | 35.31% | 32.05% | 28.27% | 26.74% |
| | PeerAiD (Jung et al., 2024) | 81.91% | 57.97% | 51.50% | 52.06% | 50.03% | 53.24% | 29.64% | 26.11% | 25.59% | 23.23% |
| | Ours (300k) | **85.09%** | **61.35%** | **55.83%** | **54.88%** | **53.21%** | **61.85%** | **36.77%** | **33.77%** | **30.08%** | **28.63%** |

Compared with the baseline ARD, our approach achieves substantial improvements in robustness, while in most cases maintaining a high level of clean accuracy. These findings indicate that our method effectively mitigates the generalization gap in adversarial distillation, enabling a more faithful transfer of the teacher model's defense capability.

Due to space limitations, we report the ablation studies A.2 and comparisons with different numbers of diffusion-based augmentations in the Appendix A.3. Our results show that augmentations selected (300k) through our screening strategy achieve better performance than simply increasing the number of augmentations (3M), highlighting the importance of augmentation quality.

## 6.2 RESULTS WITH A STRONGER TEACHER

When the teacher model is replaced with the stronger WRN-34-20, the experimental results in Table 3 reveal an interesting phenomenon: many existing distillation methods fail to gain significant benefits from the stronger teacher, and in some cases, their robust accuracy even decreases. This suggests that current approaches struggle to fully exploit the robustness of stronger teachers, leading to limited distillation effectiveness. In contrast, our method continues to show clear advantages in this setting. For student models such as ResNet-18 and MobileNet-V2, our approach achieves remarkable improvements in key robustness metrics such as PGD-20 and AutoAttack, consistently surpassing all competing methods. These results confirm that our method remains stable and superior even under the more challenging scenario of distillation from stronger teachers.

Table 3: Results with stronger teacher on CIFAR-10 dataset.

| Model | Method | Clean Acc ↑ | FGSM ↑ | PGD-20 ↑ | CW2 ↑ | AutoAttack ↑ |
|-------|--------|-------------|--------|----------|-------|--------------|
| WRN-34-20 | Teacher | 90.42% | 71.80% | 65.24% | 63.35% | 62.47% |
| RN-18 | ARD  (Goldblum et al., 2020) | 83.07% | 58.36% | 51.37% | 50.04% | 48.47% |
| | RSLAD  (Zi et al., 2021) | 85.07% | 61.10% | 53.68% | 52.20% | 50.93% |
| | IAD  (Zhu et al., 2021) | 82.38% | 58.62% | 52.04% | 50.48% | 48.91% |
| | AdaAD  (Huang et al., 2023) | 86.40% | 61.87% | 54.62% | 53.06% | 51.65% |
| | PeerAiD (Jung et al., 2024) | 85.06% | 61.01% | 54.37% | 55.25% | 52.89% |
| | Ours (300k) | **87.99%** | **66.23%** | **57.82%** | **57.50%** | **55.37%** |
| MN-V2 | ARD (Goldblum et al., 2020) | 79.47% | 55.60% | 50.77% | 48.98% | 47.45% |
| | RSLAD (Zi et al., 2021) | 83.68% | 59.14% | 53.52% | 51.36% | 50.05% |
| | IAD  (Zhu et al., 2021) | 81.34% | 57.62% | 52.36% | 49.92% | 48.36% |
| | AdaAD  (Huang et al., 2023) | 83.85% | 58.23% | 52.88% | 50.37% | 48.69% |
| | PeerAiD (Jung et al., 2024) | 81.91% | 57.97% | 51.50% | 52.06% | 50.03% |
| | Ours (300k) | **87.23%** | **64.79%** | **56.74%** | **56.40%** | **54.27%** |

## 7 CONCLUSION

This work uncovers the distinctive role of data augmentation in Adversarial Robustness Distillation and highlights its fundamental difference from standard distillation. Our analysis shows that effective augmentation must balance sample diversity and teacher reliability. This insight is also relevant to broader learning scenarios where augmentation quality is critical.

## ETHICS STATEMENT

This work adheres to the ICLR Ethics Guidelines. Our research focuses on improving adversarial robustness distillation (ARD) for lightweight models. The experiments are conducted exclusively on publicly available benchmark datasets (CIFAR-10 and CIFAR-100), which contain no sensitive, private, or personally identifiable information. No human subjects, medical data, or other personally sensitive data are involved.

We acknowledge that methods enhancing adversarial robustness could potentially be misused to build malicious models that resist detection. However, the primary motivation of this research is to strengthen the reliability and security of machine learning systems deployed in safety-critical and resource-constrained environments. We release our code and models with the intention of fostering transparency, reproducibility, and further scientific progress.

## REPRODUCIBILITY STATEMENT

For full reproducibility, we have included a detailed description of our experimental setup in the Experiment Setup section. The code has been anonymized for the review process; however, upon acceptance, we will make our complete codebase and trained models publicly available.

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

# A APPENDIX

## A.1 EXPERIMENTAL SETUP

### A.1.1 DATASET AND MODELS

This study evaluates the effectiveness of our method primarily on the benchmark image datasets CIFAR-10 and CIFAR-100 (Krizhevsky et al., 2009). We compare our approach with several representative adversarial distillation methods under different scenarios, including ARD (Goldblum et al., 2020), IAD (Zhu et al., 2021), RSLAD (Zi et al., 2021), AdaAD (Huang et al., 2023), and PeerAiD (Jung et al., 2024).

For the CIFAR-10 task, we consider two teacher models: the WideResNet-34-10 (Zagoruyko & Komodakis, 2016) pretrained model trained with TRADES (Zhang et al., 2019) provided by Zi et al. (2021), and our WideResNet-34-20 model trained with the DMIAT (Wang et al., 2023). To enhance the teacher's robustness, the latter model strictly follows the original methodology by utilizing an additional 300k generated samples (which do not overlap with the data generated in our method). For CIFAR-100, we employ the WideResNet-34-10 model pretrained with TRADES (Zhang et al., 2019). In each teacher scenario, we test the robust distillation effectiveness using two student architectures: ResNet-18 (He et al., 2016) and MobileNetV2 (Sandler et al., 2018).

### A.1.2 EVALUATION METRICS

We evaluate method effectiveness through both the classification accuracy on clean test data and the performance under adversarial attacks. Specifically, we consider several representative adversarial attacks, including FGSM (Goodfellow et al., 2014), PGD (Madry et al., 2017), $CW_\infty$ (Carlini & Wagner, 2017), and AutoAttack (Croce & Hein, 2020), as robustness evaluation metrics. The maximum perturbation is set to 8/255, with maximum iteration steps of 20 for both PGD and $CW_\infty$ attacks.

### A.1.3 IMPLEMENTATION DETAILS

All experiments are conducted with a batch size of 256. The maximum learning rate is set to 0.1 for 300 training epochs, with the learning rate schedule following the cosinew strategy. In the active selection experiments, we follow (Wang et al., 2023) and use EDM (Karras et al., 2022) to generate one million class-balanced augmented samples, from which we select a subset that remains class-balanced. We use SGD momentum optimizer. Following prior work (Wang et al., 2022), we adopt random flip and crop as the baseline augmentation, applied in all scenarios. Other methods, such as CutMix, are performed on top of these basic transformations.

## A.2 ABLATION STUDY

We conduct ablation studies on the CIFAR-10 dataset, using WRN-34-10 as the teacher model and ResNet-18 as the student model. The experiments compare performance under different selection criteria and trade-off parameters, demonstrating the effectiveness of the proposed method.

Table 4: Ablation study.

| Method | Clean Acc ↑ | FGSM ↑ | PGD-20 ↑ | CW2 ↑ | AutoAttack ↑ |
|---|---|---|---|---|---|
| $\alpha = 0$ | 84.79% | 60.80% | 54.15% | 53.10% | 51.49% |
| $\alpha = 0.3$ | 85.02% | 61.95% | 55.13% | 54.25% | 52.60% |
| $\alpha = 0.5$ | 84.83% | **62.27%** | 56.03% | **54.83%** | **53.25%** |
| $\alpha = 0.7$ | 84.85% | 61.63% | 55.87% | 54.47% | 52.93% |
| $\alpha = 1$ | **85.12%** | 62.14% | **56.13%** | 54.51% | 52.23% |

Results show that balancing data diversity and teacher fidelity yields better performance than either criterion alone, corroborating our theoretical analysis.

### A.3 EFFECT OF GENERATED SAMPLES QUANTITY ON ROBUSTNESS

In this section, we analyze the impact of augmenting the model with different quantities of randomly generated diffusion samples. Specifically, we compare the performance of the model using varying numbers of diffusion samples (100k, 300k, 500k, 1m, 3m) with the baseline method.

Table 5: Effects of varying diffusion-generated sample quantities on CIFAR-10.

| Augment Method | Clean Acc ↑ | FGSM ↑ | PGD-20 ↑ | CW2 ↑ | AutoAttack ↑ |
|---|---|---|---|---|---|
| Base | 84.59% | 60.94% | 53.41% | 52.46% | 50.58% |
| Diffusion-100k | 85.09% | 61.90% | 55.61% | 54.01% | 52.20% |
| Diffusion-300k | 85.09% | 62.05% | 55.68% | 54.27% | 52.71% |
| Diffusion-500k | _85.38%_ | 61.78% | 55.77% | 54.30% | 52.78% |
| Diffusion-1M | 84.86% | 61.42% | 55.56% | 53.79% | _53.38%_ |
| Diffusion-3M | 85.16% | _62.01%_ | _56.04%_ | _54.88%_ | 53.24% |
| Ours (300k) | **85.38%** | **62.27%** | **56.08%** | **55.23%** | **53.63%** |

As shown in Table 5, increasing the number of generated samples up to one million consistently improves robustness against adversarial attacks (FGSM, PGD-20, CW2, AutoAttack). However, further scaling to three million yields only marginal gains, with the defense success rate against AutoAttack even declining. This suggests that simply enlarging the augmentation pool does not necessarily translate into better robustness, highlighting the importance of sample quality rather than sheer quantity.

### A.4 A1

Let $h$ be the adversarial loss and $g$ satisfies Assumption 4.4. Then, for all student parameters $\theta_1, \theta_2$ and all $z \in \mathcal{Z}$, the following properties hold:

For all subgradients $d(\theta, x) \in \partial_\theta h(\theta, x)$, we have

$$\|d(\theta_1, x) - d(\theta_2, x)\| \leq L_\theta \|\theta_1 - \theta_2\| + 2(L_z L_T + L_x)\epsilon.$$

**Proof.**

$$\|\nabla_\theta g(\theta; x_1', f_T(x_1')) - \nabla_\theta g(\theta; x_2', f_T(x_2'))\|$$
$$\leq \|\nabla_\theta g(\theta; x_1', f_T(x_1')) - \nabla_\theta g(\theta; x_1', f_T(x_2'))\| + \|\nabla_\theta g(\theta; x_1', f_T(x_2')) - \nabla_\theta g(\theta; x_2', f_T(x_2'))\|$$
$$\leq L_z \|f_T(x_1') - f_T(x_2')\| + L_x \|x_1' - x_2'\|$$
$$\leq L_z (L_T \|x_1' - x_2'\|) + L_x \|x_1' - x_2'\|$$
$$= (L_z L_T + L_x) \|x_1' - x_2'\| \leq (L_z L_T + L_x) \cdot 2\epsilon$$

$$\|\nabla_\theta g(\theta_1; x_1', f_T(x_1')) - \nabla_\theta g(\theta_2; x_2', f_T(x_2'))\|$$
$$\leq \|\nabla_\theta g(\theta_1; x_1', f_T(x_1')) - \nabla_\theta g(\theta_1; x_2', f_T(x_2'))\| + \|\nabla_\theta g(\theta_1; x_2', f_T(x_2')) - \nabla_\theta g(\theta_2; x_2', f_T(x_2'))\|$$
$$\leq (L_z L_T + L_x) \|x_1' - x_2'\| + L_\theta \|\theta_1 - \theta_2\|$$

### A.5 ANALYSIS OF DIFFERENT ADVERSARIAL DISTILLATION LOSSES

Let the adversarial examples for the parameters $\theta_1$ and $\theta_2$ be:

$$x_1 \in \arg \max_{\|x' - x\|_p \leq \epsilon} g(\theta_1, x', f_T(x')),$$
$$x_2 \in \arg \max_{\|x' - x\|_p \leq \epsilon} g(\theta_2, x', f_T(x')).$$

Then we have

$$\|h(\theta_1, x, f_T(x)) - h(\theta_2, x, f_T(x))\|$$
$$= |g(\theta_1, x_1, f_T(x_1)) - g(\theta_2, x_2, f_T(x_2))|$$
$$\leq \max\{|g(\theta_1, x_1, f_T(x_1)) - g(\theta_2, x_1, f_T(x_1))|, |g(\theta_1, x_2, f_T(x_2)) - g(\theta_2, x_2, f_T(x_2))|\}$$
$$\leq L \|\theta_1 - \theta_2\|.$$

Hence, the loss function h is L-Lipschitz continuous.

However, if the adversarial generation function does not align with the core component of the loss function $h$, the L-Lipschitz continuity of $h$ with respect to $\epsilon$ may no longer hold. For instance, consider the following case:

$$x_1 \in \arg\max_{\|x'-x\|_p \leq \epsilon} g(\theta_1, x', y),$$

$$x_2 \in \arg\max_{\|x'-x\|_p \leq \epsilon} g(\theta_2, x', y).$$

In this case,

$$\|h(\theta_1, x, f_T(x)) - h(\theta_2, x, f_T(x))\|$$
$$= |g(\theta_1, x_1, f_T(x_1)) - g(\theta_2, x_2, f_T(x_2))|$$
$$= |g(\theta_1, x_1, f_T(x_1)) - g(\theta_1, x_1, y) - (g(\theta_2, x_2, f_T(x_2)) - g(\theta_2, x_2, y))) + g(\theta_1, x_1, y) - g(\theta_2, x_2, y)|$$
$$\leq \max\{|g(\theta_1, x_1, y) - g(\theta_2, x_1, y)|, |g(\theta_1, x_2, y) - g(\theta_2, x_2, y)|\}$$
$$+ |g(\theta_1, x_1, f_T(x_1)) - g(\theta_1, x_1, y) - (g(\theta_2, x_2, f_T(x_2)) - g(\theta_2, x_2, y)))|$$
$$\leq L\|\theta_1 - \theta_2\| + L_z(\||f_T(x_1) - y\|| + \||f_T(x_2) - y\||).$$

Since the term $L_z(\||f_T(x_1) - y\|| + \||f_T(x_2) - y\||)$ is independent of $\theta$, the loss function $h$ is not Lipschitz continuous. At this point, the loss function $h$ can be regarded as $\eta'$-approximately L-Lipschitz continuous, where $\eta' = L_z(\||f_T(x_1) - y\|| + \||f_T(x_2) - y\||)$.

In another scenario, the RSLAD method,

$$x_1 \in \arg\max_{\|x'-x\|_p \leq \epsilon} g(\theta_1, x', f_T(x)),$$

$$x_2 \in \arg\max_{\|x'-x\|_p \leq \epsilon} g(\theta_2, x', f_T(x)).$$

Thus,

$$\|h(\theta_1, x, f_T(x)) - h(\theta_2, x, f_T(x))\|$$
$$= |g(\theta_1, x_1, f_T(x)) - g(\theta_2, x_2, f_T(x))|$$
$$\leq \max\{|g(\theta_1, x_1, f_T(x)) - g(\theta_2, x_1, f_T(x))|, |g(\theta_1, x_2, f_T(x)) - g(\theta_2, x_2, f_T(x))|\}$$
$$\leq L\|\theta_1 - \theta_2\|.$$

This shows that $f_T(x)$ is unaffected by the $\max$ operation, and can thus be treated as an approximate hard label during the derivation. A similar case arises in the derivation of the gradient continuity for $h$. Similarly, it can be demonstrated that the function component in adversarial distillation computed from clean samples remains unaffected by adversarial perturbations. Consequently, this part of the loss influences only the continuity of the function $h$, without introducing any additional effects on the stability analysis.

## A.6 THEOREM 4.8

$$\varepsilon_{\text{gen}, \mathcal{D}_T}^{\text{Target}} = \left| \mathbb{E}_{x' \sim \mathcal{D}_T}\big[g(s(\hat{\theta}, x'), y)\big] - \mathbb{E}_{x' \sim \mathcal{D}_T}\big[g(s(\hat{\theta}, x'), f_T(x'))\big] \right|$$

$$= \left| \mathbb{E}_{x' \sim \mathcal{D}_T}\Big[g(s(\hat{\theta}, x'), y) - g(s(\hat{\theta}, x'), f_T(x'))\Big] \right| \quad \text{(linearity of expectation)}$$

$$\leq \mathbb{E}_{x' \sim \mathcal{D}_T}\Big[\big|g((\hat{\theta}, x'), y) - g(\hat{\theta}, x'), f_T(x'))\big|\Big] \quad \text{(Jensen's inequality)}$$

$$\leq \mathbb{E}_{x' \sim \mathcal{D}_T}\Big[L_x \|y - f_T(x')\|\Big] \quad \text{(by Lipschitz assumption)}$$

$$= L_x \mathbb{E}_{x' \sim \mathcal{D}_T}\Big[\|f_T(x') - y\|\Big].$$

### A.7 USE OF LARGE LANGUAGE MODELS (LLMS)

We used large language models (LLMs) in a limited capacity, solely for minor linguistic improvements like proofreading and refining the clarity of certain sections (e.g., Related Work, Experiments, and Appendix). All core scientific ideas, methodologies, and results presented in this paper are the original work of the authors. This use of LLMs was strictly for polish and does not compromise the scientific integrity of our work.

