# OpenReview forum: "Rethinking Data Augmentation for Adversarial Distillation: An Excess Risk Perspective"
_ICLR.cc/2026/Conference — Submitted to ICLR 2026_

### Official Review · Reviewer_o9WH · 2025-10-16

**Soundness:** 2
**Presentation:** 2
**Contribution:** 2
**Rating:** 2
**Confidence:** 5

**Summary:**

This paper investigates the role of data augmentation in Adversarial Distillation (AD), starting from the observation that standard augmentation techniques can degrade, rather than improve, student model robustness. To explain this phenomenon, the authors provide a theoretical analysis using an excess risk bound. Motivated by this analysis, the paper proposes a new framework, ASDA (Active Selection for Diffusion-based Augmentation). ASDA first leverages a diffusion model to generate a large pool of candidate samples and then actively selects a subset based on criteria designed to be both informative and reliable for the teacher model. The authors conduct experiments on CIFAR-10 and CIFAR-100 to demonstrate that their proposed method achieves state-of-the-art results compared to existing AD baselines.

**Strengths:**

1. The paper addresses an important and relatively under-explored research question: the role and effectiveness of data augmentation in AD.
2. The proposed method, ASDA, achieves strong empirical results, reaching state-of-the-art performance on the CIFAR-10 and CIFAR-100 benchmarks when compared against the selected AD baselines.

**Weaknesses:**

1. The introduction portrays robust overfitting as a general issue in AD largely on the basis of the vanilla ARD example; however, this characterization does not hold uniformly across AD baselines. For example, RSLAD does not exhibit the same overfitting behavior and can even improve test robustness when training is extended (e.g., to 300 epochs). Consequently, the motivation for positioning data augmentation as a necessary remedy for “robust overfitting in AD” is not fully persuasive.

2. The reported improvement from using 300k diffusion-generated samples as augmentation appears somewhat trivial and does not convincingly demonstrate the novelty of the proposed approach. Prior work [1] has already shown that diffusion-generated data can substantially enhance adversarial training robustness even without any distillation. Although the authors employ different model setups, the reported robustness levels are comparable to what standard adversarial training (e.g., TRADES) typically achieves when trained with diffusion augmentations (around 53–54% AutoAttack accuracy on ResNet-18). Therefore, it remains unclear whether ASDA provides a substantial advantage beyond what can already be achieved by straightforwardly applying diffusion-based augmentation.



3. To convincingly demonstrate the effectiveness of ASDA, the experimental design should directly compare against other adversarial training and distillation baselines under the same diffusion-augmented setting. As it stands, the reported improvement could largely reflect the benefit of diffusion augmentation itself rather than the proposed active selection mechanism. Without such cross-method comparisons, it remains unclear whether the observed advantage truly stems from ASDA’s selective augmentation strategy or simply from the use of diffusion-generated data.

4. If the authors intend to argue that diffusion-based augmentation itself enhances adversarial distillation, the paper should be reorganized to explicitly support that claim. Merely reporting performance gains does not reveal why diffusion data benefit AD. A more convincing argument would require analyzing how such augmentation affects the teacher–student relationship — for example, whether it facilitates better logit matching, improves teacher reliability under adversarial perturbations, or reduces the teacher–student output divergence. Without such mechanistic analysis, it remains unclear whether the proposed approach truly advances our understanding of how data augmentation contributes to adversarial distillation, or simply reports another case where diffusion-generated data empirically improve robustness.

5. The theoretical section closely mirrors [2] — it follows the same on-average stability framework, assumptions, and proof structure. Because of this, it reads more like a corollary than a new theorem. Furthermore, this framework fails to provide any insight into why diffusion-based augmentation is the preferred source of samples for AD. The theory only justifies the filtering component of ASDA, offering no principled justification for why diffusion models should be the source of augmentation in the first place. Therefore, this part does not warrant such a large portion of the main paper. It would be better to reduce its emphasis, strengthen the experimental analysis, or introduce a more original theoretical perspective that goes beyond existing data-dependent stability analyses.



6. The experimental comparison in Table 1 is not entirely fair because the Diffusion-300k setting introduces a substantially larger amount of training data than the other augmentation methods. While Cutout, CutMix, RandAugment, and AutoAugment operate on the same 50k CIFAR-10 samples, Diffusion-300k expands the training set by seven times through synthetic image generation. Consequently, the reported performance gain may stem largely from the increased data volume rather than the augmentation quality.

7. The Diffusion-300k setting increases the training set from 50k to ~350k samples—a 7× increase in training steps under the same schedule. Yet the paper does not report training time, GPU hours, throughput, or the cost of generating/storing diffusion samples (even generate 1M and select 300k).

From here, minor mistakes, which do not affect much on my evaluation.

8. The experimental setup focuses mainly on CIFAR-10 and CIFAR-100. Evaluations on additional datasets such as Tiny-ImageNet or SVHN are necessary to demonstrate broader generalization and verify whether the proposed method scales effectively to more complex or diverse data distributions.

9. Several reported baseline numbers deviate markedly from the original papers, casting doubt on the fairness of comparisons. For instance, on CIFAR-10 with ResNet-18, the paper reports 48.69% AutoAttack for AdaAD, whereas the original AdaAD reports 52.96% under the same dataset/architecture. Although the authors mention using different implementation practices, such discrepancies should be clearly acknowledged and explained.


[1] Wang, Zekai, et al. Better Diffusion Models Further Improve Adversarial Training. ICML 2023.

[2] Wang, Yihan, Shuang Liu, and Xiao-Shan Gao. "Data-dependent stability analysis of adversarial training." Neural Networks 183 (2025): 106983.

**Questions:**

- Can you quantify the practical benefit of the ASDA selection mechanism compared to simply using a randomly selected subset of the diffusion samples?
- Is your proposed scoring function superior to a simpler baseline, such as only selecting samples that the teacher model classifies correctly?
- It is known that diffusion-based augmentation can significantly improve standard TRADES, which does not use a teacher. While your ASDA selection mechanism requires a teacher, if a teacher is used only for the selection of the dataset, would training a standard TRADES model on these ASDA-selected samples be more effective than training TRADES on the full set of diffusion-generated samples?


Typos
- L672, L674, C&W attack label mismatch
- RandAugment (L159) and Randaugment (L79)

---

> ### Author Response · Authors · 2025-12-02
> **Response to Reviewer o9WH – Weaknesses (Part I)**
>
> We sincerely thank Reviewer o9WH for the thoughtful comments and suggestions. Below, we address each identified weakness and question in detail and describe the corresponding revisions or clarifications to be made in the updated version.
>
> ---
>
> **Weakness #1: Claiming robust overfitting as a general issue in AD is unconvincing, since it is only observed in vanilla ARD but not in stronger baselines.**
>
> We respectfully clarify that robust overfitting is not limited to vanilla ARD. In our experiments, we observe the same issue in stronger distillation baselines such as RSLAD. We will include visualizations of robust accuracy curves (e.g., AutoAttack) during training to illustrate that overfitting also occurs in RSLAD. These results will be added to the main paper.
>
> | Method      |Dataset| Best Robust Epoch |
> |-------------|-------|-------------------|
> |RSLAD   |cifar 10| 264               |
> |RSLAD   |cifar100| 222               |
>
> ---
>
> **Weakness #2: The gain from ASDA seems marginal compared to directly using diffusion-based augmentation in AT; novelty over prior work is unclear.**
>
> Thank you for raising this important point. To clarify the benefits of our method, we have conducted additional comparisons with diffusion-augmented adversarial training (Diffusion+AT) using TRADES under CIFAR-10 with ResNet-18 as the student. Results are shown below:
>
> | Method                        | Teacher       | Clean Acc | FGSM   | PGD-20 | C&W    | AutoAttack |
> |------------------------------|---------------|-----------|--------|--------|--------|-------------|
> | Diffusion+AT (random 300k)   | -             | 88.32%    | 64.35% | 56.04% | 56.04% | 53.20%      |
> | Diffusion+AT (random 1M)     | -             | 87.90%    | 64.14% | 57.60% | 56.20% | 53.50%      |
> | Ours (1M select 300k)        | wrn-34-10     | 85.38%    | 62.27% | 56.08% | 55.23% | 53.63%      |
> | Ours (1M select 300k)        | wrn-34-20     | 87.99%    | 66.23% | 57.82% | 57.50% | 55.37%      |
>
> These results demonstrate that while weak teachers yield comparable results to Diffusion+AT, stronger teachers combined with our filtering framework significantly outperform Diffusion+AT. This supports the value of distillation when coupled with principled sample selection.
>
> ---
>
> **Weakness #3: Why use diffusion-based data augmentation? Does ASDA’s benefit mainly come from the diffusion augmentation itself or the filtering mechanism?**
>
> We use diffusion-based augmentation due to its strong semantic consistency, which has been shown to benefit robustness. However, we have also conducted additional experiments applying our sample selection strategy to other augmentations (e.g., AutoAugment, CutMix, Cutout). As shown in Appendix, ASDA consistently improves robustness regardless of the base augmentation used. Furthermore, we show that selecting 300k high-quality diffusion samples from 1M outperforms using all 1M randomly, highlighting that our improvement stems from principled selection rather than sheer volume.
>
> ---
>
> **Weakness #4: Lack of Mechanistic Explanation for Why Diffusion Benefits Adversarial Distillation.**
>
> We thank the reviewer for this thoughtful question. Diffusion-based samples contribute to improved robustness in our framework for two key reasons:
>
> - **Semantic alignment**: Diffusion models generate samples that maintain semantic coherence while introducing diversity. This enables better alignment with the teacher’s feature space, which supports more stable and informative supervision under adversarial perturbations.
> - **Optimization of the diversity–reliability trade-off**: ASDA explicitly balances diversity (entropy) and reliability (teacher–label gap). Diffusion-augmented data provides a broad and semantically rich candidate pool, which allows this trade-off to be optimized effectively.
>
> While our theoretical analysis primarily focuses on scoring design, it also highlights a general mechanism: overly difficult samples—especially those where the teacher lacks confidence—can undermine robust learning. This motivates the use of reliability-aware selection and offers a conceptual framework that generalizes beyond diffusion-based augmentation.
>
> We will clarify this point in the revision to better emphasize how the proposed scoring mechanism not only leverages the properties of diffusion samples, but also provides broader guidance for future ARD methods.

---

> ### Author Response · Authors · 2025-12-02
> **Response to Reviewer o9WH – Weaknesses (Part II) and Questions**
>
> **Weakness #5: Theoretical Novelty and the Justification for Using Diffusion-Based Augmentation.**
>
> We appreciate this question. Our theoretical analysis is, to our knowledge, the first to connect teacher reliability with the generalization gap in adversarial robustness distillation. This leads to our key insight: robust distillation requires balancing sample informativeness (diversity) and supervision quality (teacher reliability).
>
> Our scoring function embodies this trade-off using entropy and teacher–label gap. While this formulation is general, diffusion-generated samples—due to their semantic consistency—provide an ideal candidate pool for optimizing this trade-off.
>
> Importantly, we show that this mechanism also improves robustness under other augmentations (e.g., AutoAugment, CutMix), confirming that the gain comes from the selection strategy, not diffusion alone. We will clarify this point in the revision.
>
> | Method (300k)         | Clean Acc | FGSM   | PGD-20 | C&W    | AutoAttack |
> |------------------------|-----------|--------|--------|--------|-------------|
> | autoaugment-random     | 84.56%    | 60.02% | 53.22% | 52.09% | 50.19%      |
> | autoaugment-select     | 84.52%    | 60.35% | 53.67% | 52.81% | 50.92%      |
> | randaugment-random     | 84.89%    | 60.62% | 53.62% | 52.43% | 50.69%      |
> | randaugment-select     | 84.50%    | 60.59% | 53.67% | 52.65% | 50.87%      |
> | cutout-random          | 84.47%    | 59.82% | 53.23% | 52.16% | 50.54%      |
> | cutout-select          | 84.72%    | 59.79% | 53.66% | 52.33% | 50.58%      |
> | cutmix-random          | 84.68%    | 60.62% | 53.84% | 52.70% | 51.08%      |
> | cutmix-select          | 84.71%    | 60.89% | 54.38% | 53.13% | 51.50%      |
>
> ---
>
> **Weakness #6-7: Is the comparison fair in terms of training data volume? Does ASDA use excessively large datasets?**
>
> Our training protocol follows prior works on diffusion-based augmentation. To avoid over-reliance on synthetic data, each training epoch samples a fixed ratio (3:7) of original to augmented data, keeping the effective training size consistent. We will clarify this protocol in the main text to ensure transparency and fairness.
>
> ---
>
> **Weakness #8: Concerns about AdaAD performance.**
>
> Thank you for pointing this out. We re-ran the experiments and corrected a reporting mistake due to checkpoint selection (we had mistakenly evaluated the checkpoint with the highest clean accuracy instead of the best robust accuracy). After correction, the results are consistent with our conclusions.
>
> ---
>
> **Question #1: Comparison with random selection.**
>
> This comparison is included in the appendix and further discussed under Weakness #3–5. Our results show that the proposed selection mechanism consistently outperforms random sampling.
>
> ---
>
> **Question #2: Comparison with teacher-correct samples only.**
>
> We conducted this comparison by selecting only samples where the teacher predicts correctly (“true-only”). Results below show that our method significantly outperforms this baseline:
>
> | Method      | Clean Acc | FGSM   | PGD-20 | C&W    | AutoAttack |
> |-------------|-----------|--------|--------|--------|-------------|
> | Teacher     | 84.92%    | 61.17% | 55.29% | 53.96% | 52.91%      |
> | True-only   | 84.64%    | 60.74% | 54.26% | 52.84% | 51.06%      |
> | Ours        | 85.38%    | 62.27% | 56.08% | 55.23% | 53.63%      |
>
> ---
>
> **Question #4: Comparison with diffusion-augmented adversarial training.**
>
> Please refer to our response to Weakness #2.
>
> ---
>
> **Performance on a larger dataset (Tiny-ImageNet)**
>
> To evaluate generalization, we added experiments on Tiny-ImageNet with ResNet-18 as the student. Results are shown below:
>
> | Method    | Clean Acc | FGSM   | PGD-20 | C&W    | AutoAttack |
> |-----------|-----------|--------|--------|--------|-------------|
> | Teacher   | 52.22%    | 28.98% | 25.58% | 24.12% | 20.79%      |
> | ARD       | 46.32%    | 23.52% | 20.42% | 19.31% | 16.31%      |
> | IAD       | 47.44%    | 26.95% | 23.65% | 22.35% | 18.95%      |
> | RSLAD     | 46.31%    | 25.45% | 22.25% | 21.05% | 17.85%      |
> | AdaAD     | 50.70%    | 27.13% | 23.83% | 22.53% | 19.03%      |
> | PeerAiD   | 55.33%    | 29.11% | 25.90% | 24.51% | 21.30%      |
> | Ours      | 52.21%    | 29.31% | 25.78% | 24.66% | 21.42%      |
>
> These results show that our method continues to offer robustness improvements on more challenging datasets, further validating its general applicability.

---

### Official Review · Reviewer_jf39 · 2025-10-27

**Soundness:** 3
**Presentation:** 3
**Contribution:** 2
**Rating:** 4
**Confidence:** 2

**Summary:**

Adversarial Robustness Distillation (ARD) is a technique for creating a robust model using a student teacher training setup. Data augmentation techniques have been used to create additional variations in the training date to improve model performance. However, in this work it is claimed that data augmentation techniques may actually hinder ARD. A new technique is proposed called Active Selection for Diffusion-based Augmentation. Empirically the results show that this technique is effective on the CIFAR-10 and CIFAR-100 datasets.

**Strengths:**

The paper is well written and easy to follow. I was not able to go through the details of all the mathematical derivations but I assume they are correct. Experimentally the method is empirically demonstrated (on CIFAR-10/100).

**Weaknesses:**

Weakness #1: There are two related works you do not cite or discuss.

1. Rethinking data augmentation:
https://www.sciencedirect.com/science/article/abs/pii/S0020025523014238
2. Another SOTA knowledge distillation approach:
https://dl.acm.org/doi/10.1016/j.procs.2025.07.114

Could you explain how these relate to your paper and/or add this into the main body?

Weakness #2: Experimental results are lacking. As a reviewer I cannot mandate that you do additional experiments. What I can say is that it would be much easier to advocate for your paper if experiments were done on Tiny-ImageNet or ImageNet or even any other non-CIFAR dataset. With only these two datasets (and CIFAR-100 essentially being a relabeling of CIFAR-10) it is hard to advocate for your paper on experimental grounds.

Weakness #3: A lot of the attacks you use are redundant or old. I appreciate that you include auto-attack as that is more SOTA, but why is so much space taken up by FGSM and PGD? I would like to see that moved to the appendix. Also, no SOTA black-box attacks (beside what is already put in AA) is used. Why are there no black-box attacks, even as simple baselines just to show how effective your method is?

Other minor points:
=Figure 1 is kind of blurry (especially the top text). Is it possible to use a higher quality figure?
=FGSM is an ancient method at this point, I would prefer if those results were removed from Table 1, it is a waste of space.
=Table 1, no epsMax value is given for the attack in the caption? This is a really important missing detail.
=I am very confused about how the T-WRA metric is actually derived. With which attack is T-WRA done?
=Table 2, please don’t report FGSM, again.
=The conclusion is extremely short and lacking in detail. If you were going to write such a sparse conclusion, why not just remove it entirely?

**Questions:**

Please answer the questions I posed in the weaknesses section. Specifically how you will handle weakness points #1, #2 and #3 and improve your paper the questions that are most important for my review.

---

> ### Author Response · Authors · 2025-12-02
> **Response to Reviewer jf39 – Weaknesses**
>
> We sincerely thank Reviewer jf39 for the constructive comments and helpful suggestions. We appreciate your feedback and address each concern below.
>
> **Weakness #1: Missing relevant citations**
>
> Thank you for pointing out these valuable references. We appreciate the opportunity to improve the completeness of our related work section.
>
> ---
>
> **Weakness #2: Lack of experimental diversity; only CIFAR-10/100 used.**
>
> We fully agree that evaluating on more diverse datasets is important to assess generalizability. In the revised version, we include experiments on the Tiny-ImageNet dataset. Our method consistently outperforms multiple strong baselines, even under this more challenging setting.
>
> | Method   | Clean Acc | PGD-20 | C&W   | AutoAttack |
> |----------|-----------|--------|-------|-------------|
> | Teacher  | 52.22%    | 25.58% | 24.12% | 20.79%      |
> | ARD      | 46.32%    | 20.42% | 19.31% | 16.31%      |
> | IAD      | 47.44%    | 23.65% | 22.35% | 18.95%      |
> | RSLAD    | 46.31%    | 22.25% | 21.05% | 17.85%      |
> | AdaAD    | 50.70%    | 23.83% | 22.53% | 19.03%      |
> | PeerAiD  | 55.33%    | 25.90% | 24.51% | 21.30%      |
> | Ours     | 52.21%    | 25.78% | 24.66% | 21.42%      |
>
> ---
>
> **Weakness #3: Too much focus on outdated attacks (FGSM, PGD); lack of black-box evaluations**
>
> Thank you for this helpful feedback. We included FGSM and PGD to ensure consistency with prior ARD literature, but we agree that FGSM is now less informative. In the revision:
>
> - We will move FGSM results to the appendix to reduce redundancy.
> - We will keep PGD and AutoAttack in the main text as standard and strong white-box evaluations.
> - For black-box attacks, AutoAttack already includes transfer-based variants. We are exploring the addition of more black-box results in the final version.
>
> ---
>
> **Other Details**
>
> Thank you for your suggestions regarding the detailed aspects of our paper. We will revise the corresponding parts in the updated version accordingly.

---

### Official Review · Reviewer_QVgB · 2025-10-29

**Soundness:** 2
**Presentation:** 2
**Contribution:** 2
**Rating:** 2
**Confidence:** 4

**Summary:**

This paper argues that augmentation methods, such as CutMix and AutoAugment, may degrade robustness transfer in logit-based distillation. To understand this problem, a comprehensive theoretical analysis is performed using excessive risk. The theoretical analysis provides a key insight: the need to balance between teacher reliability and sample diversity. Based on this analysis, the authors propose a filtering mechanism. The proposed method first generates a large pool of candidate images using a diffusion model and then applies the filtering method to select the best candidates. The filtering mechanism uses a linear combination of the entropy of the teacher's predictions and the distance between the teacher's prediction and the ground truth.

**Strengths:**

The paper is generally well written, investigates an interesting and important question, and provides a comprehensive theoretical analysis.

**Weaknesses:**

On line 154, the authors mention that there is a large generalization gap in adversarial distillation.
> In Adversarial Robustness Distillation (Goldblum et al., 2020), a large robustness generalization gap is observed: as shown in Fig. 1, although training robustness can reach above 90%, test robustness may drop to around 40%, revealing a severe failure to generalize.

However, a similar gap is also observed in adversarial training [6, 7]. Best models often achieve 40-70% robustness even when their clean accuracy is 95-99%.

Moreover, in the next line, the authors posit whether data augmentation can bridge the gap.

> This raises the question of whether data augmentation, which is effective in standard Knowledge Distillation, can mitigate such a gap.

However, the authors do not mention any prior works explaining the connection between data augmentation and generalization (the difference between train and test time error).

An important aspect of this work is the idea that augmentation may degrade adversarial distillation. However, some prior works [4] have shown a strong connection between adversarial robustness and augmentation. More importantly, prior work [2] has also shown how some augmentation can help improve robustness distillation.

Discussion on some important relevant robustness distillation  [1, 2, 3] and synthetic data for robustness [5, 8, 9] is missing.

[1] Chan, A., Tay, Y., & Ong, Y. S. (2020). What it thinks is important is important: Robustness transfers through input gradients. In Proceedings of the IEEE/CVF Conference on Computer Vision and Pattern Recognition (pp. 332-341).

[2] Awais. M., Zhou, F., Xie, C., Li, J., Bae, S. H., & Li, Z. (2021). Mixacm: Mixup-based robustness transfer via distillation of activated channel maps. Advances in neural information processing systems, 34, 4555-4569.

[3] Shao, R., Yi, J., Chen, P. Y., & Hsieh, C. J. (2021). How and when adversarial robustness transfers in knowledge distillation?. arXiv preprint arXiv:2110.12072.

[4] Zhang, L., Deng, Z., Kawaguchi, K., Ghorbani, A., & Zou, J. (2020). How does mixup help with robustness and generalization?. arXiv preprint arXiv:2010.04819.

[5] Schmidt, L., Santurkar, S., Tsipras, D., Talwar, K., & Madry, A. (2018). Adversarially robust generalization requires more data. Advances in neural information processing systems, 31.

[6] Nakkiran, P. (2019). Adversarial robustness may be at odds with simplicity. arXiv preprint arXiv:1901.00532.

[7] Zhang, H., Yu, Y., Jiao, J., Xing, E., El Ghaoui, L., & Jordan, M. (2019, May). Theoretically principled trade-off between robustness and accuracy. In International Conference on machine learning (pp. 7472-7482). PMLR.

[8] Alayrac, J. B., Uesato, J., Huang, P. S., Fawzi, A., Stanforth, R., & Kohli, P. (2019). Are labels required for improving adversarial robustness?. Advances in Neural Information Processing Systems, 32.

[9] Carmon, Y., Raghunathan, A., Schmidt, L., Duchi, J. C., & Liang, P. S. (2019). Unlabeled data improves adversarial robustness. Advances in neural information processing systems, 32.

**Questions:**

he proposed filtering method uses Entropy to make sure the samples are diverse. The entropy is maximum when all class labels are equally likely ($/dfrac{1}{K}$ for $K$ classes). Would not this be against the overall objective? In other words, the best samples are selected in such a way that the teacher would not provide useful information to the student. Correct me if I am wrong.

How do authors explain prior works showing a connection between robustness and augmentation [4], and how augmentation could be useful for robustness distillation?

How efficient is the proposed method, as an important aspect of distillation is efficiency? The active filtering requires generating an adversarial perturbation for each generated sample, and distillation also requires perturbation generation.

It would be interesting to see a sample of selected images.

On line 37, the authors mention
> Adversarial Training (AT) (Szegedy et al., 2013), currently the most practical and effective defense, trains models directly on adversarial examples to substantially improve robustness.

I think the reference is wrong. It should have been Madry et al [1].

[1] Madry, A., Makelov, A., Schmidt, L., Tsipras, D., & Vladu, A. (2018, February). Towards Deep Learning Models Resistant to Adversarial Attacks. In the International Conference on Learning Representations.

---

> ### Author Response · Authors · 2025-12-02
> **Response to Reviewer QVgB– Weaknesses**
>
> We sincerely thank Reviewer QVgB for the detailed review and valuable suggestions. We appreciate your recognition of our theoretical analysis and your thoughtful concerns regarding the generalization gap in adversarial robustness distillation (ARD). We address your comments point-by-point below.
>
> ---
>
> **Weakness #1: The generalization gap has already been studied in adversarial training (AT).**
>
> Thank you for pointing out that robustness generalization gaps are also present in adversarial training (AT). Our goal is not to claim this issue is unique to adversarial distillation (ARD), but to examine how data augmentation interacts specifically with the distillation supervision paradigm. While both AT and ARD suffer from generalization gaps, our work focuses on a failure mode particularly salient in ARD: augmentation strategies that increase diversity without considering teacher reliability may degrade student robustness.
>
> To improve clarity, we will revise the introduction and related work sections to explicitly reference works on generalization in adversarial training (e.g., [6, 7]) and better distinguish our contribution in the context of distillation.
>
> ---
>
> **Weakness #2: Lack of discussion on prior work [2], which shows that augmentation (e.g., mixup) can improve robustness distillation.**
>
> Thank you for pointing this out. We clarify that MixACM is not a method for adversarial robustness distillation. It operates entirely under standard (non-adversarial) settings, using mixup to regularize feature alignment rather than to improve adversarial robustness. In contrast, our work specifically studies augmentation under adversarial supervision, and investigates the role of teacher reliability in this process.
>
> ---
>
> **Weakness #3: Insufficient discussion of how prior work (e.g., [4]) connects data augmentation and generalization.**
>
> We appreciate this feedback. Zhang et al. [4] discusses how mixup improves generalization in natural training by smoothing the loss surface. However, their work does not explore adversarial training or robustness distillation, and the attacks used (e.g., FGSM) are relatively weak. While our paper already discusses augmentation and generalization in AT (Sections 2.2 and 2.3), we agree it would be valuable to better clarify these differences.
>
> ---
>
> **Weakness #4: Missing citations to related work on robustness distillation and augmentation-based AT.**
>
> Thank you for the suggestion. We will include the cited related works in our revised version to strengthen the contextual foundation of our study.

---

> ### Author Response · Authors · 2025-12-02
> **Response to Reviewer QVgB – Questions**
>
> **Question #1: Why select high-entropy samples? Don’t they risk providing noisy supervision?**
>
> Great question. In our framework, entropy is used as a proxy for promoting sample diversity, which we consider essential for alleviating the generalization gap in adversarial robustness distillation (ARD). As widely observed in data augmentation research, increasing sample diversity helps improve generalization. Low-entropy samples often correspond to highly confident teacher predictions and carry limited information, making them less helpful for student training. In contrast, moderately high-entropy samples tend to be more informative and better expose the student model to challenging decision boundaries.
>
> We also understand and appreciate your concern. Extremely high entropy may indeed indicate that the teacher model is uncertain, which compromises the quality of supervision. Our theoretical analysis supports this intuition: when difficult samples exceed the teacher's capacity to provide reliable guidance, distillation may fail to benefit.
>
> To address this, we introduce teacher–label distance as an explicit reliability constraint in the scoring function. This complements the entropy term and forms a joint diversity–reliability trade-off. The design ensures that selected samples are both informative and teachable, leading to more effective robustness distillation.
>
> ---
>
> **Question #2: How do the authors explain prior works showing a connection between robustness and augmentation [4], and how augmentation could be useful for robustness distillation?**
>
> We acknowledge that Zhang et al. [4] provides valuable insights into how Mixup can improve robustness and generalization in natural training settings. However, we would like to clarify that their work does not study Mixup in the context of adversarial training (AT) or robustness distillation. Their results focus on standard (non-adversarial) training with Mixup, and demonstrate improved resistance to simple attacks such as FGSM, primarily due to the smoothing of the loss landscape and regularization of input gradients. These findings do not directly extend to adversarial settings such as AT or ARD, where the supervision dynamics and optimization challenges differ significantly.
>
> ---
>
> **Question #3: How efficient is the proposed method, given that both filtering and distillation involve adversarial perturbations?**
>
> Thank you for raising this important question. To control computational cost, we design the sample filtering process as a one-time pre-processing step performed before training, rather than repeatedly during distillation. We believe this additional cost remains within an acceptable range and is justified by the significant robustness improvements observed in our experiments.
>
> ---
>
> **Question #4: Citation error (Szegedy et al.)**
>
> We thank the reviewer for pointing out the citation error. We will correct the reference to Madry et al. [1] in the revision.

---

### Official Review · Reviewer_3imA · 2025-11-01

**Soundness:** 3
**Presentation:** 2
**Contribution:** 3
**Rating:** 4
**Confidence:** 4

**Summary:**

This paper addresses why traditional data augmentation methods such as CutMix and AutoAugment, which have been known to be effective in standard Knowledge Distillation (KD), degrade performance in Adversarial Robustness Distillation (ARD). Through theoretical analysis grounded in uniform stability and excess risk bounds, the authors reveal a trade-off between augmentation diversity and teacher reliability in ARD, and then propose Active Selection for Diffusion-based Augmentation (ASDA) that leverages diffusion-generated samples and selects teacher-reliable, informative augmentations using entropy and teacher output fidelity.

**Strengths:**

- Interesting theoretical analysis: The excess-risk–based generalization analysis for ARD is new and offers valuable insight into the impact of data augmentation. The proposed “diversity–reliability trade-off” is intuitive and theoretically supported.
- Clear empirical motivation: The observation that common augmentations hurt ARD is well documented with empirical evidence (Table 1).
- Combining diffusion augmentation with active selection is conceptually simple yet effective.

**Weaknesses:**

- Simplistic selection mechanism: The active selection simply relies on a linear combination of entropy and teacher output distance, which seems heuristic and may not generalize beyond diverse data.

- Lack of ablation on teacher reliability: The claim that low teacher reliability increases generalization error is theoretically grounded but empirically underexplored.

- Insufficient evaluation dataset: CIFAR-10/100 are very small benchmarks. Large-scale or real-world evaluation would strengthen the claims.

- Limited comparative study: Baselines are restricted to ARD-style methods; there is no direct comparison to modern diffusion-augmented adversarial training or other data selection frameworks.

**Questions:**

- Could the method generalize to non-diffusion-based data or real-image augmentations?
- Can the authors provide a quantitative correlation between teacher reliability metrics (T-WRA, T-OS) and student generalization?
- Would integrating the selection process into joint training (rather than pre-selection) lead to better performance?

---

> ### Author Response · Authors · 2025-12-02
> **Response to Reviewer 3imA – Weaknesses**
>
> We sincerely thank Reviewer 3imA for the constructive feedback. We are glad that you found our theoretical analysis interesting and appreciated the simplicity and effectiveness of our diffusion-based selection strategy.
>
> ---
>
> **Weakness #1: Is the proposed selection strategy (entropy + teacher distance) too simplistic?**
>
> We understand your concern about the simplicity of our sample selection function. Our goal is not to introduce complexity for its own sake, but to isolate two fundamental factors we found to be critical in adversarial robustness distillation (ARD): sample diversity (captured via entropy) and teacher reliability (measured by the gap between teacher predictions and the ground-truth labels).
>
> To validate this design, we conducted an ablation in Appendix by varying the weighting coefficient α in the selection score. In particular, setting α=0 uses only teacher-label distance, while α=1 uses only entropy. Both of these single-factor variants consistently underperform the combined setting on CIFAR-10 under FGSM, CW2, and AutoAttack, confirming the necessity of balancing both diversity and reliability.
>
> In the revision, we will move this key ablation to the main paper, clarify the role of α, and include visual comparisons of selected samples under different α values to improve interpretability.
>
> ---
>
> **Weakness #2: Lack of ablation on teacher reliability**
>
> Thank you for pointing this out. Our method is designed to balance two key factors: sample diversity and teacher reliability, using a unified scoring function. Since these components interact closely, it is nontrivial to isolate one without influencing the other.
>
> To provide further insight, we analyze how changing the weighting coefficient α in the selection score affects both sample diversity and teacher reliability. Specifically, we track how the teacher's reliability changes with α by monitoring T-WRA (teacher’s worst-case robust accuracy during training). We observe that when α increases, meaning the selection process emphasizes diversity while downplaying reliability, T-WRA drops significantly, and the student’s robustness also degrades.
>
> | α     | AutoAttack | T-WRA   |
> |-------|-------------|----------|
> | 0     | 51.49%      | 93.97%   |
> | 0.3   | 52.60%      | 93.40%   |
> | 0.5   | 53.25%      | 80.28%   |
> | 0.7   | 52.93%      | 70.25%   |
> | 1     | 52.23%      | 64.31%   |
>
> ---
>
> **Weakness #3: Insufficient evaluation dataset**
>
> We fully agree that evaluating on more diverse datasets is important to assess generalizability. In the revised version, we include experiments on the Tiny-ImageNet dataset. Our method consistently outperforms multiple strong baselines, even under this more challenging setting.
>
> | Method   | Clean Acc | FGSM  | PGD-20 | C&W   | AutoAttack |
> |----------|-----------|-------|--------|-------|-------------|
> | Teacher  | 52.22%    | 28.98%| 25.58% | 24.12%| 20.79%      |
> | ARD      | 46.32%    | 23.52%| 20.42% | 19.31%| 16.31%      |
> | IAD      | 47.44%    | 26.95%| 23.65% | 22.35%| 18.95%      |
> | RSLAD    | 46.31%    | 25.45%| 22.25% | 21.05%| 17.85%      |
> | AdaAD    | 50.70%    | 27.13%| 23.83% | 22.53%| 19.03%      |
> | PeerAiD  | 55.33%    | 29.11%| 25.90% | 24.51%| 21.30%      |
> | Ours     | 52.21%    | 29.31%| 25.78% | 24.66%| 21.42%      |
>
> ---
>
> **Weakness #4: Limited comparative study (e.g., TRADES)**
>
> Thank you for the suggestion. Since our focus is on adversarial robustness distillation, we chose standard ARD pipelines for comparison to ensure controlled evaluation. That said, we agree that comparing with diffusion-augmented adversarial training is valuable and will include such experiments in the revision.

---

> ### Author Response · Authors · 2025-12-02
> **Response to Reviewer 3imA – Questions**
>
> **Question #1: Could the method generalize to non-diffusion-based data or real-image augmentations?**
>
> Thank you for pointing this out. While we primarily adopt diffusion-based augmentations due to their strong performance on general robustness benchmarks, our proposed sample selection strategy is agnostic to the augmentation type.
>
> To evaluate this, we are running additional experiments using non-diffusion augmentations such as CutMix and AutoAugment, as well as real-image perturbations. One challenge in applying real images from unrelated distributions is that teacher reliability scores become hard to compute due to label mismatch. A possible extension we are considering is to select real samples that are semantically close to each target class and assign them pseudo-labels accordingly, enabling evaluation of selection under mild distribution shifts.
>
> Preliminary results show consistent improvement over random selection, confirming the generality of our approach. These results will be added to the revised submission.
>
> | Method (300k)        | Clean Acc | FGSM  | PGD-20 | C&W   | AutoAttack |
> |----------------------|------------|--------|--------|--------|-------------|
> | AutoAug-Random       | 84.56%     | 60.02% | 53.22% | 52.09% | 50.19%      |
> | AutoAug-Select       | 84.52%     | 60.35% | 53.67% | 52.81% | 50.92%      |
> | RandAug-Random       | 84.89%     | 60.62% | 53.62% | 52.43% | 50.69%      |
> | RandAug-Select       | 84.50%     | 60.59% | 53.67% | 52.65% | 50.87%      |
> | Cutout-Random        | 84.47%     | 59.82% | 53.23% | 52.16% | 50.54%      |
> | Cutout-Select        | 84.72%     | 59.79% | 53.66% | 52.33% | 50.58%      |
> | CutMix-Random        | 84.68%     | 60.62% | 53.84% | 52.70% | 51.08%      |
> | CutMix-Select        | 84.71%     | 60.89% | 54.38% | 53.13% | 51.50%      |
>
> ---
>
> **Question #2: Can the authors provide a quantitative correlation between teacher reliability metrics and student generalization?**
>
> This is an insightful question, and we agree that understanding the link between teacher reliability and student performance is important. As addressed in Weakness #2, our ablation over α shows that reducing emphasis on teacher reliability (larger α) leads to lower T-WRA and weaker student robustness. This confirms the correlation and supports our scoring design.
>
> ---
>
> **Question #3: Would integrating the selection process into joint training (rather than pre-selection) lead to better performance?**
>
> Thank you for raising this important question. We agree that integrating the selection process into joint training is a promising direction.
>
> However, our current goal is to isolate and understand the trade-off between sample diversity and teacher reliability in adversarial robustness distillation. A pre-selection framework allows us to study this trade-off more cleanly, without introducing additional dynamics during training.
>
> Despite its simplicity, our method achieves strong performance across multiple benchmarks, supporting the effectiveness of the proposed scoring design. We consider extending the selection to a dynamic, training-integrated strategy as a valuable direction for future work.

---

### Meta-Review · Area_Chair_pppR · 2026-01-10

**Summary:**

The authors investigate the negative impact of standard data augmentation on Adversarial Robustness Distillation (ARD), noting that techniques like CutMix can reduce student model robustness. They propose ASDA, a method that uses diffusion-generated samples and a scoring function to balance sample diversity and teacher reliability. While the theoretical motivation regarding excess risk was recognized as interesting, the reviewers expressed significant concerns regarding the incremental novelty over existing diffusion-based adversarial training, the simplicity/heuristic nature of the selection mechanism, and the computational overhead required to generate and filter 1M samples for marginal gains.

**Reviewer Concerns:**

Addressed in Rebuttal
- Evaluation on Diverse Datasets: The authors provided new results on Tiny-ImageNet, showing that ASDA outperforms several ARD baselines.
- Comparison with Diffusion+AT: To address concerns that the benefit came solely from diffusion data, the authors added a comparison with TRADES+Diffusion; ASDA (55.37% AA) outperformed the random diffusion baseline (53.50%) when using a strong teacher.
- Generality to other Augmentations: The authors demonstrated that their selection scoring provides consistent (though often small) improvements when applied to CutMix and AutoAugment.

Outstanding Concerns
- Incremental Novelty and Theoretical Depth: Reviewer 09WH noted the theory mirrors existing data-dependent stability frameworks and acts more as a corollary than a new theorem. The justification for specifically using diffusion models beyond them being a "high-quality candidate pool" remains theoretically thin.
- Efficiency and Practicality: The process requires generating 1 million diffusion samples and performing adversarial filtering (via FGSM) before distillation even begins. Reviewers remain concerned that the training time, GPU hours, and storage costs are not adequately reported or justified by the resulting performance increases.
- Selection Heuristics: The scoring function remains a simple linear combination of entropy and label distance. Reviewers felt this was a heuristic approach that lacks a deeper mechanistic analysis of why this specific balance is optimal across different architectures.
- Discrepancy in Baselines: Concerns regarding reported baseline numbers (e.g., AdaAD) were partially addressed by a "reporting mistake" correction, but this raised further questions about the rigor of the initial evaluation.

**Reviewer Scores:**

- Reviewer QVgB: 2-> 2
While citations were corrected, the reviewer's fundamental skepticism about the "diversity-reliability" scoring as a contribution remains largely unswayed.
- Reviewer o9WH: 2-> 2
Despite the new Diffusion+AT comparison , this reviewer's critique of the theoretical section being unoriginal and the lack of reported training costs  remains a major barrier.
- Reviewer jf39: 4-> 4
The added Tiny-ImageNet and clarified evaluation protocol address the main concerns. However, the reviewer does not say much about the strength of the paper and does not demonstrate appreciation for the technical contributions made in this paper.
- Reviewer 3imA: 4->4
Likely appreciative of Tiny-ImageNet results but still noted the selection mechanism is simplistic

---

### Decision · Program_Chairs · 2026-01-26

Reject